# Assessing Growth-Promoting Activity of Bacteria Isolated from Municipal Waste Compost on *Solanum lycopersicum* L.

**Pallavi Bhardwaj** [1], **Abhishek Chauhan** [2], **Anuj Ranjan** [3], **Saglara S. Mandzhieva** [4], **Tatiana Minkina** [3], **Usha Mina** [5], **Vishnu D. Rajput** [3,*] and **Ashutosh Tripathi** [1,*]

1   Amity Institute of Environmental Sciences, Amity University, Noida 201303, Gautam Budh Nagar, India
2   Amity Institute of Environmental Toxicology, Safety and Management, Amity University,
    Noida 201303, Gautam Budh Nagar, India
3   Academy of Biology and Biotechnology, Southern Federal University, Stachki 194/1,
    344090 Rostov-on-Don, Russia
4   Kalmyk Scientific Center of the Russian Academy of Sciences, str. I.K. Ilishkina, 8, 358000 Elista, Russia
5   School of Environmental Sciences, Jawaharlal Nehru University, New Delhi 110067, India
*   Correspondence: rvishnu@sfedu.ru (V.D.R.); atripathi1@amity.edu or tripathiashutos@gmail.com (A.T.)

**Abstract:** Rapid urbanization and population growth are stressing the present agricultural systems and could threaten food security in the near future. Sustainable development in agriculture is a way out to such enormously growing food demand. Plant growth-promoting bacteria (PGPB) are considered pivotal to providing adequate nutrition and health to plants and maintaining soil microbial dynamics. In the present study, municipal solid waste composts (MSWC) were studied for the presence of PGPB and their growth-promoting characteristics such as ammonium production, siderophores production, phosphorus solubilization and potassium solubilization, IAA (indole acetic acid), and HCN production. Four promising isolates were chosen and identified through 16S rRNA sequencing as *Bacillus* sp. strain L5-1, *Bacillus pumilus* strain EE107-P5, *Bacillus* sp. strain LSRBMoFPIKRGCFTRI6 and *Bacillus* sp. strain LPOC3. The potential of isolates is validated using *Solanum lycopersicum* (tomato) and was found to improve its growth significantly. The findings indicated the presence of potential *Bacillus* strains in MSWCs, and these composts can be utilized as biofertilizers for urban agricultural practices. However, studies concerning their impact on other crops' growth and health are still underdeveloped. Since MSWCs might carry hazardous metals or chemicals, their evaluation for the safe application on the crops should also be assessed.

**Keywords:** urban waste; agriculture; culturable fraction; microbes; plant growth potential; bacterial diversity; nutrients

## 1. Introduction

The Food and Agriculture Organization (FAO), United Nations has reported that by 2050, the global population will reach 9.1 billion and demand for grain crops will increase by 70% [1]. The ever-growing population is certainly stressing the limited natural resources. Apart from this, industrialization leads to constant soil degradation, climate change, and water scarcity, affecting global food security [2]. It also leaves several chemical footprints in the soil and water that ultimately affect food safety. The overutilization of fertilizers and pesticides affects environmental ecosystems including soil, water, non-target organism, and human health [3,4]. Sustainable development in agriculture technology has the potential to help ameliorate and strengthen such issues and restore the deteriorating health of the ecosystem. It is well established that the soil bacteriome significantly supplies nutrients to crops, stimulates plant growth via phytohormones, inhibits/controls plant infections and pests, and improves soil conditions by nutrient solubilization, nitrogen fixation, combined with pollutant mineralization [5–8]. The interactions among plants and microbes in the rhizosphere were sought for modification, mobilization, and volatilizing nutrients from a

restricted nutrient pool, their successive uptake, and to indicate their genetic adaptability and potential [9].

Plant Growth Promoting Bacteria (PGPB) have been diversely studied to boost the growth of plants and for biocontrol which is gaining attention in modern agriculture for improving crop health and yield [10]. They improve growth through direct or indirect mechanisms. These bacteria also have phyto-stimulant capabilities, as well as the potential to engage in biogeochemical cycles actively [11,12]. PGPB, with the potential to solubilize phosphate (P) from the rock and other inorganic forms, are crucial for P availability in the soil. The PGPB with nitrogen ($N_2$) fixation ability helps make N available in the rhizosphere; it helps in improving the root length and density impact on systemic plant metabolism; and microbial phyto-protection [13]. They do produce compounds like non-ribosomal peptides, and polyketides that are effective in the biocontrol of plant pathogens, optimal health of the plants and improved yield [14].

The ongoing drift in population and urbanization is creating another significant environmental issue i.e., waste management. It has been observed that waste from the kitchen and sewage increases the presence of organic materials in municipal solid waste (MSW). Composting MSW is an innovative approach to recycling organic materials and acquiring a stable, nutrient-rich product [15]. The qualities of compost are influenced by microbial availability and dynamics. The presence of potential PGPB in compost would impart additional values to the compost. The PGPB in bio-oxidative composting can produce stable and humic-rich compounds which support plant development and phytopathogen control [16]. Inoculating agricultural fields with compost improves soil microbiota and physical qualities, lowering production costs and increasing crop yield [17]. Many symbiotic and non-symbiotic bacteria present in compost or soil are being used for crop yield and disease management all over the world [18].

It has been discovered that the microbial extract of composts is useful in fostering plant growth and limiting fungal infections [19]. *B. cereus*, *Arthrobacter woluwensis*, *Paenarthrobacter nitroguajacolicus*, and *B. mycoides* were discovered to have higher PGP activity than other bacteria detected in the soil [20]. *B. pumilus* has been observed to increase the growth of beans [21], rice [22], and mustard with radish [23]. It also enhances the tomato quality owing to its ability to boost nutrient assimilation by encouraging root development [24]. In comparison to other PGPB, inoculating lettuce plants with *B. pumilus* significantly boosted head weight, height and vigor [25].

Several studies have reported and extensively studied the PGPB and have explored its commercial interests. However, there is very little data about the characterization of PGPB isolated from MSWC. Therefore, the objective of the present study aimed to use culture-dependent techniques to characterize microbes isolated from MSWC for their PGP traits. With this, the study has explored the effects of characterized PGPB on the growth of *Solanum lycopersicum* (tomato). The purpose of using tomato as part of this study is that it transforms quite easily, it is frequently employed as a model crop for studies on various physiological, cellular, biochemical, molecular, and genetic processes as well as the development of fruits. It is simple to grow in growth chambers or greenhouses [26].

## 2. Materials and Methods

### 2.1. Compost Sample Collection Sites

There are four landfills with three Waste to Energy (WtE) plants in Delhi, India. The compost samples were collected from two different WtE companies stated below.

1. IL&FS Environmental Infrastructure & Services Ltd. (IEISL): It has been a unified waste management organization since 2007. The compost plant at Okhla is operated and maintained by IL & FS Waste Management and Urban Services Ltd. (IWMUSL).
2. Delhi MSW Solutions Limited (DMSWSL): DMSWSL is a unit of Hyderabad-based Ramky Enviro Engineers Ltd. and is located in Bawana Industrial Area.

## 2.2. Preparation of Compost Samples

Freshly prepared compost samples were collected from Bawana and Okhla plants, New Delhi and labeled with reference codes BC and OC signifying the sampling site and the date of sampling was mentioned. Collected samples were immediately stored in the refrigerator at 4 °C for further processing in the laboratory. The samples were homogenized and a total of 30 subsamples (15 × 2) from each bag were prepared aseptically and stored in sterile sealed polypropylene bags. These prepared samples were further processed for microbiological studies and physicochemical analysis.

## 2.3. Physicochemical Properties of MSWC

Compost samples were evaluated for estimation of organic carbon (OC), nitrogen (N), available phosphorus (P), potassium (K), and pH. The Walkley and Black approach were used to assess OC [27]. The available N, available P and available K were estimated using the Kjeldahl method [28], Bray's P-1 method [29] and ammonium acetate ($C_2H_7NO_2$) extraction method, respectively [30]. The pH of the compost samples was analyzed with a digital pH measuring device (Labman, LMPH-10) by making compost to water suspension in 1:2 (*w/v*) [31].

## 2.4. Isolation and Characterization of Culturable Bacteria in the Compost

The bacterial isolates were cultured using the pour-plate approach on the nutrient agar (NA) medium followed by 10-fold dilutions. For this, 10 g of the sample was diluted in 90 mL of 0.9% saline solution. Distinct agar mediums were used in order to assess the PGP potential of the isolates. The spread-plate cultures were sealed with parafilm and incubated at 25 ± 1 °C in the dark till bacterial colonies appeared. Each dilution was replicated three times. Then, the colonies were differentiated morphologically, isolated and refined by sub-culturing them on fresh agar plates using the streak plate method. Each isolate was assigned a unique code, and further subculturing was performed to isolate only pure colonies. Pure isolates were grown on NA slants at 4 °C prior to before using them in experiments and for long-term conservation, were cryo-preserved at −20 °C in 50% (*v/v*) glycerol.

## 2.5. Quantitative Estimation of Bacteria

For the enumeration of bacterial population per gram of compost, 1 mL sample from each prepared dilution was collected and seeded onto NA medium plates to carry out multiple streak isolation. The number of colonies based on the morphology was counted using a digital colony counter. Equation (1) was used to count the number of colonies (CFU $g^{-1}$) present in compost [32].

$$N = \frac{\sum C}{(n_1 + 0.1n_2)d} \times \frac{10}{weight\ of\ sample\ taken} \tag{1}$$

where, $N$ is the sum of colony-forming units in one gram of compost, $d$ stands for dilution factor, from which the primary counts were attained, $n_1$ and $n_2$ refer to the number of plates considered in the first and second dilution, and $\sum C$ indicates the sum of all colonies counted on all the plates.

## 2.6. Inoculum Preparation

The selected isolates after inoculating into an aseptic nutrient broth were kept under incubation at 30 °C for 48 h in a shaker incubator at 150 rpm [20]. Centrifugation was done at 5000 rpm for 10 min to separate the cells. The cell pellets were resuspended in sterilized normal saline for achieving an optical density of 1.0 with ~7 × $10^8$ cells $mL^{-1}$. These prepared cultures were further used as 1.0% (*v/v*) inoculum to investigate PGP attributes.

### 2.7. Screening of Pure Bacteria Isolates for Assessment of PGPB Traits

The bacterial isolates are screened for different PGP attributes including P- solubilization, K-solubilization, $NH_3$ production, IAA production, HCN production, Siderophore production and N-fixing ability using standard methods.

### 2.7.1. Determining P-Solubilizing Activity of Isolates

P-solubilizing activity depends upon the emergence of evident halo zones around the bacterial colonies capable of solubilizing calcium phosphate. Following the standard procedure by Mehta and Nautiyal, 2001 [33], P-solubilization was examined qualitatively on Pikovskaya medium (PVK). Using the media plates, three replicates of each bacterial isolate were created. The plates were then sealed with parafilm, kept under incubation at 30 °C and examined for the formation of halo zones after 7 days. These produced zones formed around colonies were further measured and recorded using a ruler, and P-Solubilization Index (PSI) was determined using Equation (2).

$$PSI\ (\%)\ =\ \frac{H - B}{B} \times 100 \qquad (2)$$

where $H$ = diameter of emerged halo zone, $B$ = diameter of the bacterial colony (cm).

### 2.7.2. Determining K-Solubilizing Activity of Isolates

The bacterial ability to solubilize K was analyzed on Aleksandrow (AKW) agar plates [34]. Overnight developed bacterial culture was streaked on agar plates and incubated at 30 ± 2 °C for 5 days. K-solubilizing efficiency was demonstrated by a clear zone surrounding the colony. K-Solubilization Index (KSI) was determined using Equation (3).

$$KSI\ (\%)\ =\ \frac{H - B}{B} \times 100 \qquad (3)$$

where $H$ = diameter of produced halo zone, $B$ = diameter of the bacterial colony (cm).

### 2.7.3. Estimation of $NH_3$ and HCN Production by Isolates

All the isolates were grown in Asbhy's N-free liquid medium for 24 h at 37 °C. The cultures were further streaked and incubated on Ashby's N-free agar plates at 37 °C for 24 h. The method proposed by Goswami et al., in 2014 was used to observe $NH_3$ production by isolates [35]. Cultures grown in liquid medium were further centrifuged for 10 min at 3000 g, and 0.2 mL supernatant was added to 1 mL Nessler's reagent, with the final volume up to 8.5 mL by adding $NH_3$-free distilled water (d.w.). The change in color from brown to yellow indicated $NH_3$ production and was measured using a spectrophotometer at 450 nm by creating a standard curve using 0.1–10 µmol of ammonium sulfate. Test tubes with no change in color were considered negative.

The bacterial ability to produce HCN was investigated following the procedure described by Lahlali et al., 2020 [36]. The bacteria were grown in solid Luria Bertani (LB) medium accompanied with 4.4 $gL^{-1}$ glycine. Each box's lid was lined with Whatman paper saturated with alkaline picrate. The dishes were parafilmed and kept under incubation for four days at 30 °C. Production of HCN was confirmed by the presence of a red-orange shade.

### 2.7.4. Estimation of IAA Production by Isolates

The Salkowski reagent method was used to test the generation of IAA by bacterial isolates [37]. The experiment used two groups of 2 mL supernatants of every isolate cultured in LB broth (one with tryptophan or one without tryptophan). Then, 1 mL Salkowski reagent was added to each broth and incubated for 30 min in the dark. The emergence of a pink tint in the broth confirmed IAA production and further absorbance was measured at 530 nm. To calculate IAA production, a calibration curve using IAA (5.0–50.0 $gmL^{-1}$) was prepared.

### 2.7.5. Estimation of Siderophore Production by Isolates

Chrome Azurol S (CAS) agar plate method was used to measure siderophore production [38]. The 48-h-aged cultures were identified on CAS agar separately and incubated for 48 h at 28 ± 1 °C. The CAS-shuttle test was used to estimate siderophores quantitatively [39]. Bacteria were cultured in LB broth for 72 h at 28 °C and 150 rpm with continual shaking. A similar volume of CAS-suttler solution was put into cell-free broth supernatants [40]. The siderophores production is shown by the absence of the blue color. The changing color of the supernatant from yellowish to orange was assessed using a spectrophotometer (630 nm). The percentage of siderophores production by each bacterial isolate was calculated using Equation (4).

$$Siderophores\ (\%)\ =\ \frac{(A_R - A_s)}{A_R} * 100 \tag{4}$$

where, $A_R$ and $A_s$ are the absorbance of reference and sample respectively.

### 2.8. Molecular Characterization of the Isolates

#### 2.8.1. DNA Extraction

Following the manufacturer's procedure, genomic DNA was obtained using the GenElute$^{TM}$ Bacterial Genomic DNA kit (Sigma-Aldrich, USA). Extracted DNA samples were quantified by taking spectrophotometric readings (OD) at 260 nm using a spectrophotometer (BioSpectrometer, Eppendorf, Germany). An OD value equal to one relates to nearly 50 μg/mL DNA (double-stranded). The concentration (purity) of DNA samples was assessed based on the ratio between OD at 260 nm and OD at 280 nm. A sample having pure DNA has a ratio between 1.8 and 2.0 [41].

#### 2.8.2. 16s rRNA Amplification of Desired Genomic DNA

PCR was used to amplify the 16S rRNA sequence using two universal primers, 27F(5′-AGAGTTTGATCCTGGCTCAG-3′) and 1492R(5′CGGTTACCTTGTTACGACTT-3′) [42]. PCR was conducted in PCR reaction tubes by adding 18 μL of master mix with a 2 μL DNA template. DNA was amplified in a thermocycler (C1000 Touch$^{TM}$ Thermal cycler, CFX96$^{TM}$ Real-Time System, Bio-Rad, made in Singapore). Denaturation (5 min at 94 °C) was followed by 30 cycles of denaturation (45 s at 94 °C), annealing (45 s at 55 °C), and extension (1 min 45 s at 72 °C).

#### 2.8.3. Phylogenetic Analysis

The strains' genetic sequences were determined Using NCBI's (National Center for Biotechnology Information) BLAST software available freely at https://blast.ncbi.nlm.nih.gov/ (accessed on 1 December 2022) and a similarity search was done to find the closest sequences. The evolutionary history of the potent isolates was inferred using the Neighbor-Joining method [43]. The bootstrap consensus tree inferred from 1000 replicates is taken to represent the evolutionary history of the taxa analyzed [44]. Branches corresponding to partitions reproduced in less than 50% of bootstrap replicates are collapsed. The percentage of replicate trees in which the associated taxa clustered together in the bootstrap test (1000 replicates) are shown next to the branches [44]. The evolutionary distances were computed using the Maximum Composite Likelihood method [45] and are in the units of the number of base substitutions per site. This analysis involved 10 nucleotide sequences. Codon positions included were 1st + 2nd + 3rd + Noncoding. All ambiguous positions were removed for each sequence pair (pairwise deletion option). There was a total of 692 positions in the final dataset. Evolutionary analyses were conducted in MEGA11 [46]. With this, the reported sequences with accession numbers were submitted to the NCBI gene bank.

## 3. Evaluating Potential Effects of Isolates on the Growth of *S. lycopersicum*

### 3.1. Seed Procurement

The seeds of *S. lycopersicum* were procured from National Seeds Corporation, Indian Agriculture Research Institute, Pusa, New Delhi, India and were tested for germination potential. The seeds were sanitized for one minute with 70% ethanol (*v/v*), then for 20 min with 2.5% of sodium hypochlorite with three washes with sterilized d.w. Floating seeds were thrown away [47].

### 3.2. Effect of Isolates on the Seed Germination Rate

Sterilized and squared filter paper was set in Petri dishes and filled with 45 seeds followed by the addition of 5 mL sterilized d.w and 0.1 mL of each isolate. Petri plates were further sealed and kept at $20 \pm 5\,°C$ for 20 days. Uninoculated seeds were used as a control in a petri dish. Every day, the total number of germinated seeds was counted, and the germination rate (GR) was determined using Equation (5) [48] after 20 days.

$$Germination\ rate\ (\%) \ = \ \frac{Germinated\ Seeds}{Total\ Seeds} \times 100 \tag{5}$$

### 3.3. Plant Material and Growth Conditions

The soil used as the substrate for the growth experiment was collected from an organic farm at the university premises. The soil was dried and filtered through with a 2 mm sized sieve before autoclaving it at $121\,°C$ for 20 min [49]. The basic chemical properties of soil were determined after autoclaving: pH (8.23), EC (0.10 ds/m), Total Nitrogen (0.14%), Organic Carbon (1.37%), C:N (9.78), available P ($7.10$ mg kg$^{-1}$), available K ($175$ mg kg$^{-1}$) and available N ($47$ mg kg$^{-1}$).

Seeds were planted in plastic nursery socket trays filled with organic medium. All the seedlings were pulled up after 3 weeks, rinsed with d.w, and the roots were drenched for 4–5 h in a solution without or with isolates ($10^8$ CFU mL$^{-1}$). Following that, every pot was stuffed with sterilized soil and prepared for isolate treatments. The treatments prepared are represented in Table 1.

**Table 1.** Prepared treatments to observe the effects of isolates' inoculation of *S. lycopersicum*.

| Treatment | Description |
|-----------|-------------|
| T-0 | Control Soil |
| T-1 | Soil inoculated with isolate BC-1 |
| T-2 | Soil inoculated with isolate BC-4 |
| T-3 | Soil inoculated with isolate OC-3 |
| T-4 | Soil inoculated with isolate OC-7 |

One uprooted and soaked seedling was planted into each pot. During plantation, 1 mL of sterile water containing isolates was added to planting openings. Moreover, seedlings were added to the pots without inoculation in the control . Fresh bacterial cultures were prepared as described previously [49,50]. Bacterial cultures were prepared in LB after achieving the optical density (1.0) of the suspension by using a spectrophotometer (600 nm). Following that, all cultures were incubated overnight in a shaker incubator ($30 \pm 2\,°C$). The cultures were centrifuged at 1610 g for 20 min to obtain pellets. At the end, the pellets were diluted with sterile d.w and inoculated.

During the experiment, the mean day/night temperature was $35/24\,°C$, with a photoperiod of 12–13 h and relative humidity of 70–72%. This experiment had four treatments and one control group with three replications (three pots) employing a completely randomized univariate layout.

*3.4. Inoculation Effects on Growth Parameters*

The growth response of plants, including plant height (PH), leaf number (LN), chlorophyll content (TC), shoot dry weight (SDW) and fresh weight (SFW) and roots (RFW and RDW) was observed. Length measurements and LN count were carried out using a meter ruler and manual counting. TC was estimated spectrophotometrically. A standard weighing balance was used to observe variations in weight (SDW, SFW, RDW, RFW) at the harvesting stage. The leaf from the top of the matured plants was harvested to analyze photosynthetic pigment [51]. A leaf sample of 0.2 g discs was mixed homogeneously with 50 mL of 80% acetone. After this, the sample was centrifuged at $15,000 \times g$ for 10 min to get the supernatant. The absorbance of the supernatant was observed at 652 nm using UV Visible Spectrometer (Perkin Elmer/Lambda 25). TC (mg/g) was calculated by using Equation (6) below.

$$Total\ Chlorophyll\ (\mathrm{mg/g\ FW})\ =\ \left(\frac{A652}{34.5}\right) * \left(\frac{1000V}{1000W}\right) \tag{6}$$

where, $A$ is the absorbance of supernatant, $V$ is the volume of sample (mL), and $W$ is the fresh weight of the sample (g)

*3.5. Harvesting and Plant Analysis*

Plants were harvested after six weeks of treatment initiation. Shoots from each pot were oven-dried to a constant weight at 65 °C. Then, roots were collected, and gently washed with water to remove soil, rinsed with deionized water, and oven-dried at 65 °C. SFW, RFW and SDW, RDW were determined before and after drying.

*3.6. Statistical Analysis*

All the obtained results were statistically analyzed using IBM SPSS statistics 23. An Independent t-test was performed to detect the significant differences between the physiochemical properties of both composts. The (ANOVA) Tukey's tests (T's-Test) were used for multiple comparisons between isolates and their effects on plants. The germination results were compared to the control group. The data has been represented in the form of graphs using Origin 2022.

## 4. Results

*4.1. Compost Physiochemical Analysis*

The pH of BC was valued at almost neutral (7.21), whereas the pH of the OC was slightly basic (7.89). The OC content for BC and OC was $14.50 \pm 1.29\%$ and $14.66 \pm 0.63\%$ respectively. The values for available N were observed at $37 \pm 1.73$ mg kg$^{-1}$ and $34 \pm 1.69$ mg kg$^{-1}$ for BC and OC respectively. Available P for BC ($271 \pm 1$ mg kg$^{-1}$) was significantly more than OC ($265 \pm 1.07$ mg kg$^{-1}$) at t (4) = 7.093, $p$ =0.002 and available K for BC ($157 \pm 2.64$ mg kg$^{-1}$) was significantly higher from OC ($143 \pm 1.32$ mg kg$^{-1}$) at, t (4) = 8.197, $p$ = 0.001. Available P and available K content were found more in BC. The results were computed as Mean $\pm$ SD and each sample had three replicates.

*4.2. Quantifying Bacteria*

Based on a calculation using Equation (1), it was observed that BC had more cultural bacteria as compared to OC. The number of culturable bacteria in BC and OC was found to be $2.8 \times 10^5$ CFUg$^{-1}$ and $2.3 \times 10^5$ CFUg$^{-1}$ respectively. Thus, it can be concluded that BC is quantitatively richer in microbial count may be due to the high concentration of heavy metals and antimicrobial substances (pesticides, paints, dyes, disinfectants etc.) in OC. The presence of such contaminants might hinder the occurrence and metabolism of microbes in OC.

### 4.3. Screening of Isolates for PGP Traits

A total of 13 bacterial isolates were successfully isolated from the compost samples (BC-1, BC-2, BC-3, BC-4, BC-5, BC-6, OC-1, OC-2, OC-3, OC-4, OC-5, OC-6 and OC-7) using continuous subculture. 4 isolates with a maximum number of PGP traits were analyzed with 16s sequencing (BC-1, BC-4, OC-3, and OC-7).

#### 4.3.1. P-Solubilizing Potential of Isolates

Results revealed that five isolates (BC-1, BC-4, OC-2, OC-5, OC-6) were capable of solubilizing P to different extents (Figure 1A). The qualitative efficiency of the isolates was significantly different from each other. BC-1 and OC-2 with maximum PSI showed the highest qualitative efficiency. PSI of BC-1 and OC-2 was 85.23% and 71.26%, respectively followed by the OC-5 exhibiting a good solubilization index of 45.66%. BC-4 with 31.53%, and OC-6 recorded the lowest PSI (25.22%).

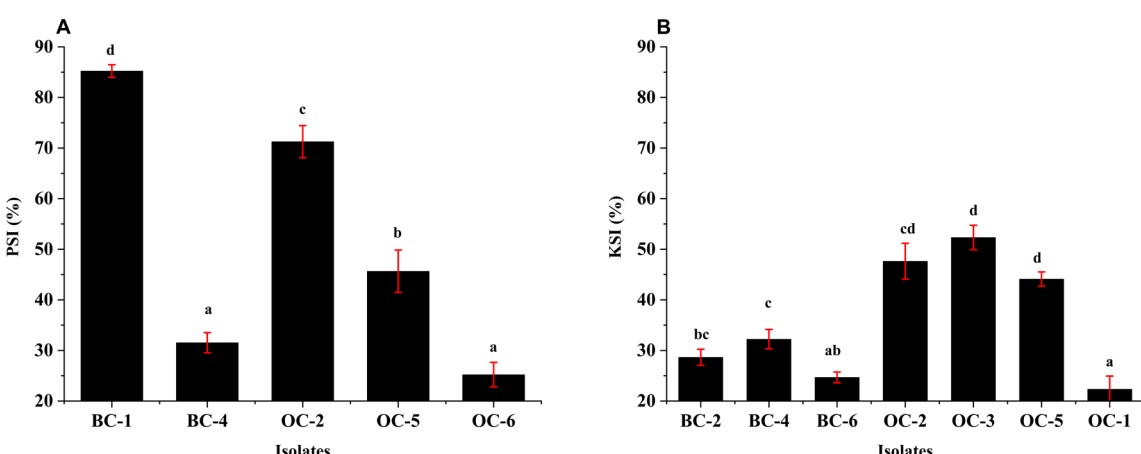

**Figure 1.** Quantitative estimation of (**A**) Phosphate Solubilizing index and (**B**) Potassium Solubilizing index of the potential Isolates. Results are expressed as means of three replicates and vertical bars shown in the figure represent standard deviations and different letters indicate significant differences at $p < 0.05$ by T's-Test.

#### 4.3.2. K-Solubilizing Potential of Isolates

Among all isolates, OC-3, OC-2, OC-5, BC-6, BC-4, and BC-2 showed K-solubilizing potential (Figure 1B). The strains OC-3, OC-2, and OC-5 had the greatest ability to solubilize K, as evidenced by their highest KSI (52.33, 47.63, and 44.12%, respectively), followed by BC-6 (38.55%). However, when compared to other strains, BC-4 (32.23%), BC-2 (28.65%), and OC-1 (22.34%) solubilized the least amount of K.

#### 4.3.3. NH$_3$ and HCN Production Potential of Isolates

It was observed that BC-2 (12.43 µmol mL$^{-1}$), OC-7 (9.87 µmol mL$^{-1}$), and BC-4 (8.25 µmol mL$^{-1}$) were producing NH$_3$ most efficiently when compared to other strains, followed by BC-1 (7.56 µmol mL$^{-1}$). The isolate BC-5 (6 µmol mL$^{-1}$) was the least efficient among all (Figure 2A). With this, all the isolates were positive for HCN production except OC-1, BC-5, OC-4, OC-2, and BC-2.

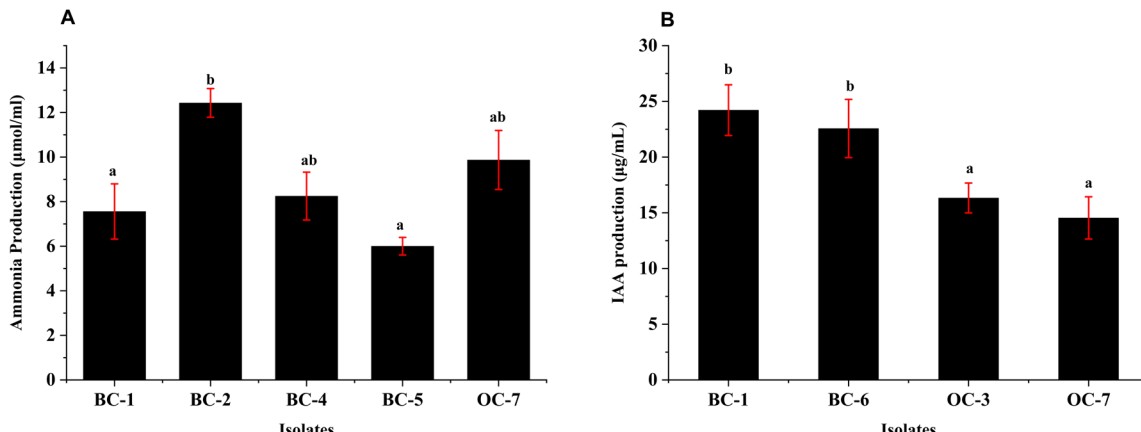

**Figure 2.** Quantitative estimation of (**A**) Ammonia Production and (**B**) Indole Acetic acid production of the potential Isolates. Results are expressed as means of three replicates and vertical bars shown in the figure represent standard deviations of the means. Significant differences among treatments explained by different alphabetical letters according to T's-test at $p \leq 0.05$.

### 4.3.4. IAA Production Potential of Isolates

Based on biochemical screening BC-1 showed the maximum production of IAA followed by BC-6, OC-3 and OC-7 with 24.21, 22.56, 16.33 and 14.54 µgmL$^{-1}$ respectively (Figure 2B). Only four isolates were found positive for IAA production.

### 4.3.5. Siderophore Production Potential of Isolates

Similarly, the CAS assay revealed that BC-1 (54%) produced the maximum number of siderophores followed by OC-3 (47%), OC-7 (43.56%), OC-2 (37.20%) and OC-1 (11.87%) was found to produce the least percentage of siderophores (Figure 3).

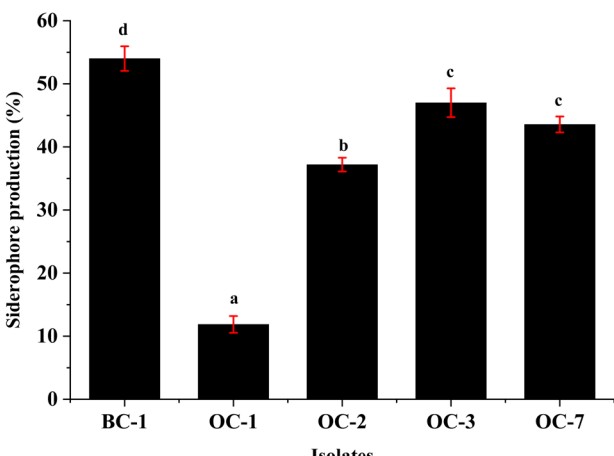

**Figure 3.** Quantitative estimation of Siderophore production by the potential Isolates. Results are expressed as means of three replicates and vertical bars shown in the figure represent standard deviations of the means. Significant differences among treatments explained by different alphabetical letters according to T's-test at $p \leq 0.05$.

### 4.4. Molecular Identification of the Potent PGPB

Four isolates that revealed the maximum potential for PGP activity were further genetically characterized (BC-1, BC-4, OC-3, and OC-7) using 16S rRNA sequence analysis Figure 4. Based on the identity of 99.71% for the 16S rRNA gene sequence, BC-1 isolate was identified as similar to *Bacillus* sp. strain L5-1 (MN784421.1), BC-4 isolate was identified as similar to *Bacillus pumilus* strain EE107-P5 (MN581181.1), OC-3 similar to *Bacillus* sp. strain LSRBMoFPIKRGCFTRI6 (MN882712.1) and OC-7 similar to *Bacillus* sp. strain

LPOC3 (MH412687.1). A phylogenetic tree of the identified isolates is shown in Figure 5 demonstrating the relationship between these four isolates stating that *Bacillus* sp. strain L5-1 and *Bacillus* sp. strain LSRBMoFPIKRGCFTRI6 are closely related whereas *Bacillus pumilus* strain EE107-P5 and *Bacillus* sp. strain LPOC3 are close species.

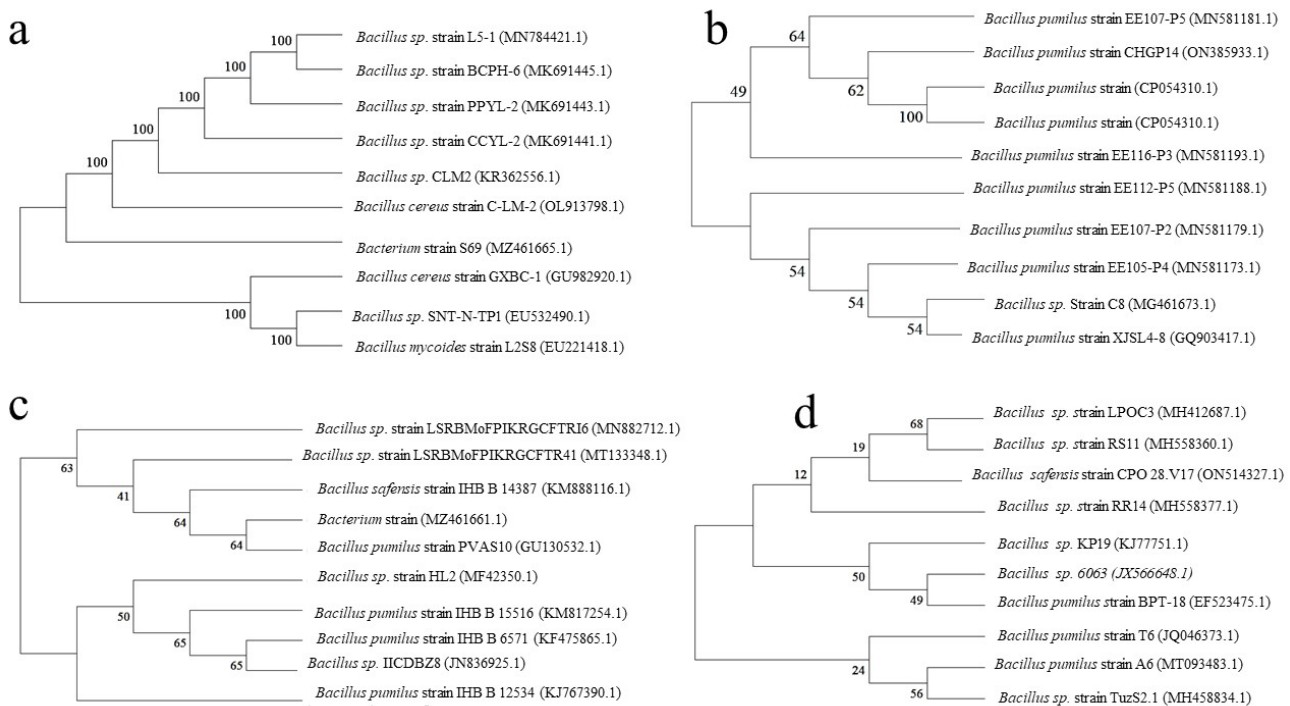

**Figure 4.** Neighbour-joining phylogenetic tree of bacterial isolates and their closely matching homologues from NCBI BLAST results. (**a**) BC-1, (**b**) BC-4, (**c**) OC-3 and (**d**) OC-7.

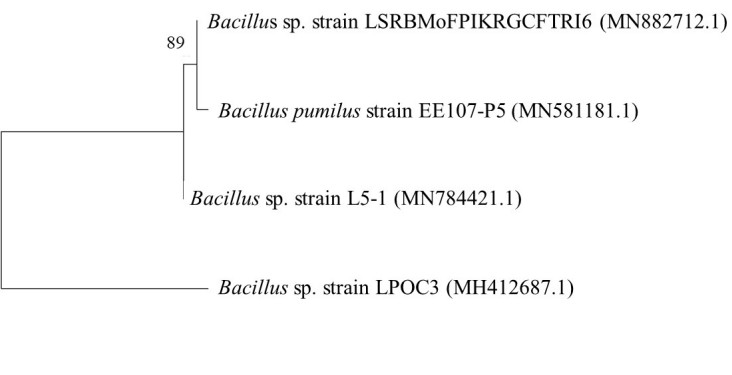

**Figure 5.** Relationship between all identified four isolates based on the phylogenetic analysis using the Neighbor-Joining method conducted in MEGA11.

Proteobacteria, Firmicutes, Bacteroidetes, and Actinobacteria are among the bacterial phyla found in composting, though at varying rates. Bacteria such as *Lactobacillus*, *Acetobacter* and *Weissella* are mesophilic organic acid producers who control the initial phases of composting [52]. On the other hand, in the successive thermophilic stage *Bacillus* spp. and Actinobacteria dominate the process and are indicators of the proper composting process [53,54]. This could be the reason for obtaining isolates belonging to *Bacillus* genera in the present study.

### 4.5. Plant Growth Influenced by Inoculation of Isolates

Our findings demonstrated that the inoculation of isolates had an impact on PH, LN, TC, SFW, SDW, RFWandRDW of *S. lycopersicum*. Improved PH was observed as a factor of isolate inoculation in the soil. It increased significantly ($p < 0.05$) by 18.5%, 11.11%, 22.22% and 29.62% by the inoculation of BC-1, BC-4, OC-3, and OC-7, respectively, as compared to control (Figure 6). The maximum increase in PH was shown by isolate OC-7 followed by OC-3. Similar findings were reported by Masood et al., where *B. pumilus* improved the PH, SFW, and chlorophyll contents when inoculated to N-amended soil [55].

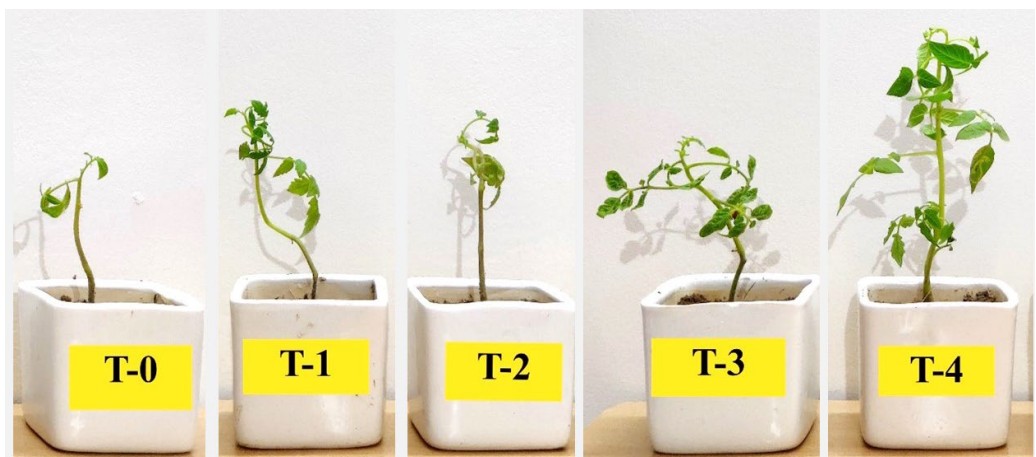

**Figure 6.** The visible effect of isolates inoculation on the shoot height of *S. lycopersicum* as compared to control.

With this, LN as a visual parameter of growth significantly improved with the addition of isolates. LN were increased by 42.85%, 34.28%, 22.85% and 20% by inoculating with OC-7, BC-4, OC-3 and BC-1 respectively as compared to the control (Figure 7). The maximum increase was observed by the inoculation of OC-7 and BC-4. On the contrary, the inoculation did not affect the TC in leaves up to a significant level. Only, inoculation with BC-1 significantly ($p < 0.05$) improved TC as compared to the control. The results were found to be similar to the study where inoculation with *B. pumilus* had no significant impact on chlorophyll content. Maximum GR i.e., 85%, was attained by the inoculation of OC-7 followed by BC-1 (80%). A study conducted by Martinez et al., 2021 showed a similar observation where germination of *S. lycopersicum* was increased by 11.12% when inoculated with *B. pumilus* after attaining the GR of 87.65% [47]. The GR was found to be 86.7% when inoculated with *B. subtilis*, and the *B. subtilis* strain GIBI 200 decreased the seed mortality in *S. lycopersicum* [56].

SFW and RFW were found to be positively influenced after inoculation (Figure 8). SFW increased by 46.66%, 33.3%, 22.2% and 11.1% by inoculation with OC-7, BC-1, BC-4 and OC-3 respectively. In comparison with control, RFW significantly increased by 34.54%, 30.90%, 16.36% and 9.09% with inoculation of OC-7, OC-3, BC-1, and BC-4 respectively. Similarly, inoculation with *Bacillus* sp. (except *Bacillus* sp. RF-37) increased root and shoot weights with increased shoot lengths, but the root lengths were observed unaffected [57]. On the other hand, SDW and RDW remained unaffected by the application of the inoculation. Analogous findings were reported by Ricci et al. 2019 [58], where leaf area and RDW were suppressed by the presence of *Bacillus* sp. in *S. lycopersicum*.

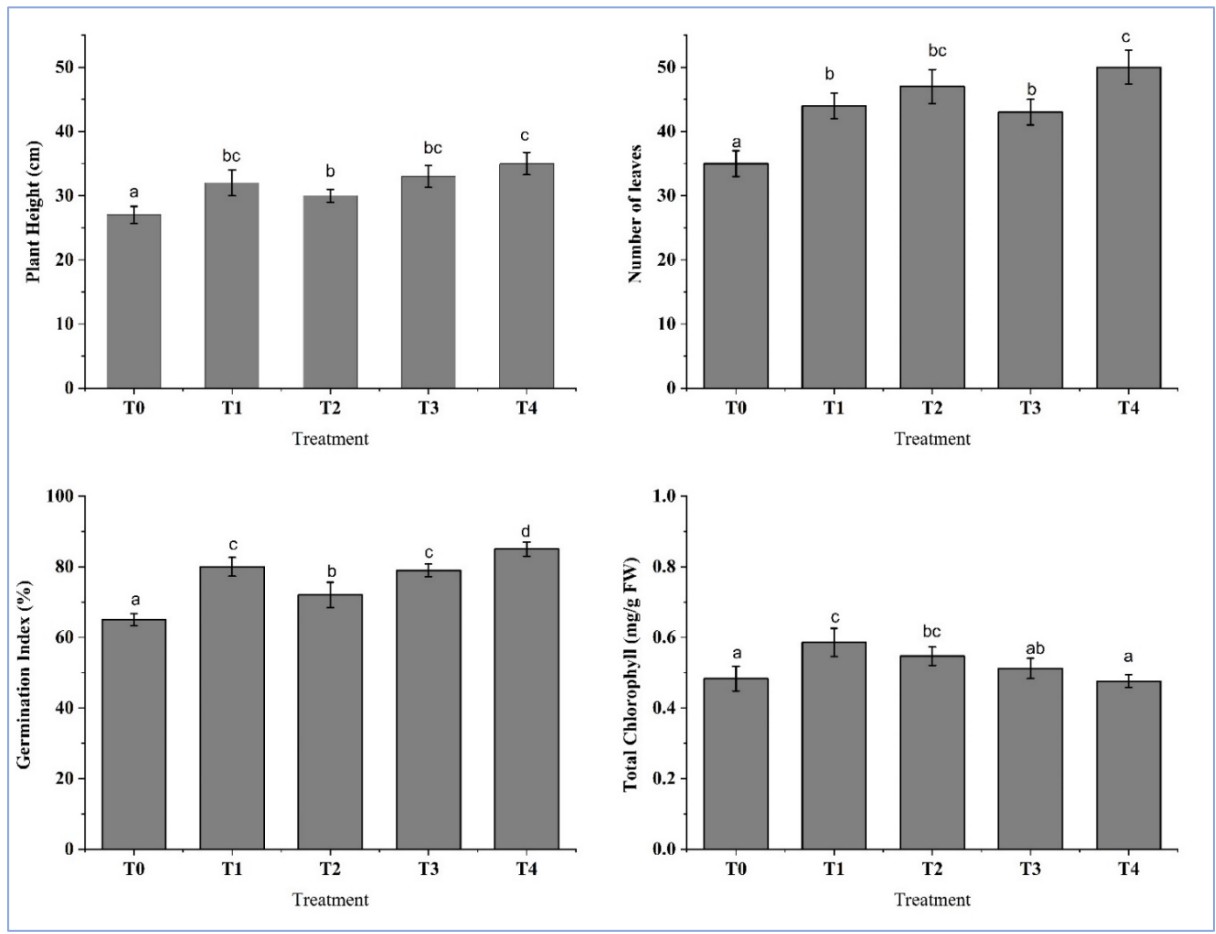

**Figure 7.** Effects of isolates inoculation on plant growth including PH, LN, GR, and TC content in *S. lycopersicum* plants. Error bars represent Means ± S.D of three replicates. Different letters above columns indicate statistically significant differences at $p \leq 0.05$.

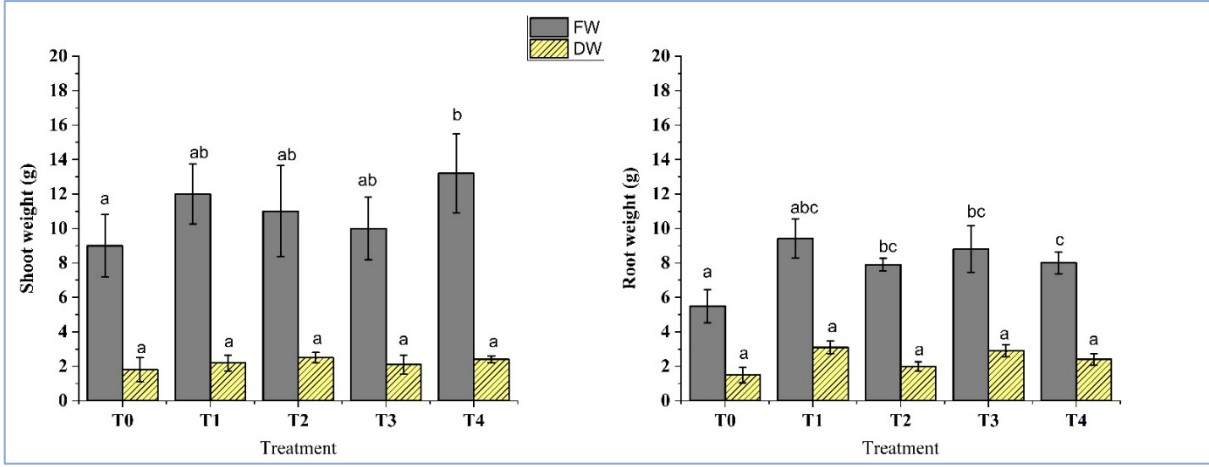

**Figure 8.** Effects of isolates inoculation on SFW, SDW, RFW and RDW of plants. Error bars represent Means ± S.D of three replicates. Different letters above columns indicate statistically significant differences at $p \leq 0.05$.

## 5. Discussion

Solid waste generation in cities and urban areas is very common. However, its recycling and utilization significantly matter for sustainable development. MSW has been a resource for several waste recycling sectors. However, the agriculture sector has hardly focused on MSW utilization. The MSW recycling can be directed towards composting which can be an effective nutrient source for crops and an alternative to chemical fertilizers [53]. Compost is known to have high agronomic value because it increases soil organic matter content (SOM), enhances water retention capacity of soils, reduces soil degradation, and bioremediate polluted soil. The presence of PGPB in soil improves crop growth by producing plant hormones, increasing nutrient availability, and improving plants' overall quality and yield [59].

Several studies have been conducted to isolate and screen effective PGPB from the soil, rhizospheres, and other environments. However, there are no significant studies that have been done on the isolation of PGPB from MSWC. In our study, we isolated PGPB from the MSWC and characterized them using 16S rRNA gene sequence analysis. The isolates were found to belong to the genera *Bacillus*. In addition to this, in-vitro analysis was done to explore the PGP traits by monitoring the development of *S. lycopersicum*.

Several species of *Bacillus* are known to bear PGP traits. *B. subtilis*, *B. pumilus* and *B. cereus* promote plant growth through the production of phytohormones such as IAA, gibberlic acid, ACC deaminase, chitosanase, protease, glucanase, cellulase, HCN, biofilm preparation and lipopeptides [60–62]. *Bacillus* facilitates the production of organic acids in soil for releasing available P to plants and increasing the activity of acid phosphatases. The study conducted by Saeid et al. 2018, confirmed that *B. megaterium*, *B. Cereus* and *B. subtilis* improved P solubilization by producing organic acids (gluconic acid, succinc acid, lactic acid, acetic acid, and propionic acids) [63].

*Acidithiobacillus ferrooxidans*, *B. mucilaginosus*, *B. circulans*, *B. edaphicus*, and *Paenibacillus* spp. have all been reported to release K in accessible forms in soils [64]. *Bacillus* species adapt to secrete organic acids like acetic, fumaric, citric, gluconic, tartaric and oxalic acids to solubilize refractory minerals like feldspar and illite to liberate K [65]. *B. subtilis* and *B. licheniformis* are reported to release organic acids (oxalic, malic, and benzoic acid) in improving the growth of *S. lycopersicum* [66].

Approximately 95% of *Bacillus* species are known to produce $NH_3$ followed by *Pseudomonas*, *Rhizobium* and *Azotobacter* by 94.2%, 74.2% and 45%, respectively [67]. A recent report stated that out of a total of 39 *Bacillus* isolates, 71.8% of the population produced $NH_3$ [68]. In present study, isolates (BC-1, BC-2, BC-4, BC-5, OC-6, and OC-7) demonstrated significant PGP properties in vitro tests among which BC-1, BC-4 and OC-7 were potent $NH_3$ producers.

Under alkaline conditions, *B. megaterium* is known to have the highest iron-chelating capacity followed by *B. subtilis* [69]. These findings demonstrate the capability of *Bacillus* strains to produce siderophores and BC-1, OC-3, and OC-7 in the present study produced siderophores at high rates. HCN has the potential to control plant bacterial diseases by inhibiting disease development. According to some studies, it does not act directly as a bio controller but rather participates in geochemical processes in the substrate such as metal chelation, which indirectly boosts P availability and increases plant growth [70,71]. BC-1, BC-3, BC-6, OC- 3, OC-5, OC-6, OC-7, and OC-8 were found to be HCN- producing, which might have enhanced the plant growth when inoculated in the soils.

A Number of *Bacillus* sp. strains have demonstrated antibacterial and antifungal activity against phytopathogens. These strains are known to have a great potential for the synthesis of a variety of volatile organic compounds (VOCs) and bioactive secondary metabolites (BSMs). These metabolites inhibit phytopathogens due to their direct antagonistic nature and induced systematic resistance (ISR), making the plant less susceptible to infection [72,73]. For example, *B. amyloliquefaciens* strains S1 and Ba01 inhibited the growth of pathogens such as *Clavibacter michiganensis* and *Streptomyces scabies* [74,75]. *B. subtilis* CAS 15 promoted the growth of Piper nigrum by ISR to *Fusarium* wilt and improved

average fruit weight and yielded up to 36.92% and 49.68%, respectively [76]. Similarly, *B. pumilus* prevented fungal growth (*Fusarium oxysporum*), and improved the GR of *Solanum lycopersicum* seeds [47].

Since the MSWCs are rich in organic waste hence harbor a variety of bacteria. The latest study on PGPB from MSWC was also done by Tondello et al., 2022 and reported the presence of *Bacillus* along with other species [54]. He characterized the isolates using 16s RNA and assessed their PGP traits including siderophore and auxin production, P-solubilization and peptide mineralization to $NH_3$. It was observed that MSWC was dominant in *B. cereus*, *B. licheniformis*, *B. thuringiensis*. Even a study conducted in Egypt, revealed that *B. licheniformis*, and *B. altitudinis* dominated commercial compost by 38 and 14%, respectively [77]. Along with it, *Bacillus* dominated the herbal vermicompost [78] and *B. flexus*, *B. cereus* and *B. subtilis* have been isolated from the poultry manure compost [79].

In the present study, all the isolates improved the growth of plants in terms of PH, LN, TC, SFW, SDW, RFW and RDW. Among four identified PGPB, OC-7 showed improved height, LN, SFW and RFW with the maximum number of positive PGP traits ($NH_3$, HCN, IAA, siderophores). These strains were examined for their PGP potential in pots to recommend them as a possible bio-fertilizer for the farmers for sustainable agriculture practices.

According to the findings, selected isolates can act as a potent plant growth promoter and have PGP properties for *S. lycopersicum*. Thus, these isolates can be used in combination or alone as biofertilizers for enhancing the quality and growth of crops. Whereas, the ability of these strains under stressed conditions and diseased conditions has not been studied yet which makes this study restricted to sterile soil conditions and to pot farming practices. This aspect of the study within combination or alone with chemical fertilizers also needs to be explored to obtain a better yield of *S. lycopersicum* and will be performed further to support the obtained information.

## 6. Conclusions

In the present work, the isolated bacteria from MSWC were screened for PGP traits and further used as inoculum on the *S. lycopersicum*. Improved plant growth inferred the presence of potential PGPB in the MSWC (dominantly *Bacillus* strains). Hence it can be effectively used as compost or in soil amendments for urban agricultural practices.

Further studies are required to investigate the presence of obnoxious elements or compounds that may accumulate in the edible part of the plants by the application of MSWC. A well-planned study is of utmost necessity that can lead to the development of value-added compost from the MSW. An approach for utilizing MSWC as a biofertilizer would be truly helpful in solid waste management in urban and cities, thereby reducing the stress on municipal corporations. Further studies related to the effect of different metal concentrations, pH, carbon source and temperature on these isolates can be helpful in identifying the mechanism adapted by them to support plant growth. This will help establish the use of these isolates as bioinoculants in different types of soils (acidic, alkaline, metal-contaminated, tropical or temperate etc.).

**Author Contributions:** Conceptualization, P.B.; methodology, P.B.; software, P.B.; validation, A.R., A.T., A.C.; formal analysis, P.B.; investigation, P.B., A.C.; resources, A.T., S.S.M., T.M., V.D.R.; data curation, A.R., A.C.; writing—original draft preparation, P.B.; writing—review and editing, P.B., A.R., A.C.; visualization P.B., A.T.; supervision, A.T., U.M., A.C.; project administration, A.T. All authors have read and agreed to the published version of the manuscript.

**Funding:** This research was funded by the Science and Engineering Research Board, Government of India, grant number "EMR/2017/004448".

**Institutional Review Board Statement:** Not applicable.

**Informed Consent Statement:** Not applicable.

**Data Availability Statement:** The data presented in this study are available on request from the corresponding author. The data are not publicly available due to privacy.

**Acknowledgments:** The authors are highly thankful to Amity University, Noida for allowing us to conduct the study and Science and Engineering Research Board (SERB), Department of Science and Technology (DST), Government of India, New Delhi, India (Grant No. EMR/2017/004488) for providing the financial assistance to perform the same.

**Conflicts of Interest:** The authors declare no conflict of interest.

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
