# Peer review of "Assessing Growth-Promoting Activity of Bacteria Isolated from Municipal Waste Compost on Solanum lycopersicum L."

_horticulturae, doi:10.3390/horticulturae9020214_

Round 1

Reviewer 1 Report

In line 23, P and K should present as phosphorus and potassium, because this was the first time they appeared in the article.

Line 26, sp., not italic

In the abstract, an overview of the experiment in this study was missing. Moreover, Solanum lycopersicum should be go along with its common name, tomato.

The reason why tomato was used as a subject to PGPB was not emphasized.

In line 246, it should be BLAST, instead of blast.

Why the BC-1 strain was chosen, while it was unable to solubilize K?

Figure 4, accession number, strains, not italic

Lines 522-524, size should uniform, scientific name should italic

Line 545, dot

Discussion should concentrate on Bacillus

Conclusion should shorten to focus the objectives

Reviewer 2 Report

Please see the attached review report.

Reviewer 3 Report

-       This study mentioned the isolation of PBPB from the municipal waste compost, and the assessment of PGPB toward S. lycopersicum L. has proceeded.

-       Interestingly, municipal waste compost can be used as a source for eco-friendly agricultural microbial agents, especially isolation sources of PGP bacteria. Also, this kind of approach coincides with the carbon-neutral policy famous worldwide.

-       However, there are a lot of things that need to change. Please refer comments below.

-       This article was submitted to the special issue “New Research of Physiological of Horticultural Crop Resistance to Abiotic Stresses.” However, the plant growth test was conducted using organic soil from the university farm. Thereby, it is considered that there is nothing to cause abiotic stress during the plant pot experiment.

-       There are no results of whole genome analysis of all isolates. Even if plant growth-promoting activities were measured by chemical assay, such as nutrient solubilization, auxin production, etc., these activities need to be confirmed based on the genomic approach to confirm PGP activity data. Furthermore, whole genome data of all isolates can provide more information about the characteristics of all isolates in this study. If it can be included in this manuscript, the characteristics of all isolates in this study can be investigated comprehensively.

(example - PGP activity related genes of Bacillus sp. BC-1.)

(example - comparison of PGP activity related genes between all isolates in this study.)

Round 2

Reviewer 3 Report

I understand that genetic research for PGPB in this study will be conducted whenever funding resources are secured.

But there is something that needs to improve for this manuscript. Actually, I uploaded this zip file in 1st revision, but there was might some problem during upload. Please refer zip file which includes word and pdf files.

Round 3

Reviewer 3 Report

Thank you for your kind reply. I hope that you have a wonderful further study.